# GENERALIZABLE NON-LINE-OF-SIGHT IMAGING WITH LEARNABLE PHYSICAL PRIORS

## ABSTRACT

Non-line-of-sight (NLOS) imaging, recovering the hidden volume from indirect reflections, has attracted increasing attention due to its potential applications. Despite promising results, existing NLOS reconstruction approaches are constrained by the reliance on empirical physical priors, e.g., single fixed path compensation. Moreover, these approaches still possess limited generalization ability, particularly when dealing with scenes at a low signal-to-noise ratio (SNR). To overcome the above problems, we introduce a novel learning-based approach, comprising two key designs: Learnable Path Compensation (LPC) and Adaptive Phasor Field (APF). The LPC applies tailored path compensation coefficients to adapt to different objects in the scene, effectively reducing light wave attenuation, especially in distant regions. Meanwhile, the APF learns the precise Gaussian window of the illumination function for the phasor field, dynamically selecting the relevant spectrum band of the transient measurement. Experimental validations demonstrate that our proposed approach, only trained on synthetic data, exhibits the capability to seamlessly generalize across various real-world datasets captured by different imaging systems and characterized by low SNRs.

## 1 INTRODUCTION

Non-line-of-sight (NLOS) imaging represents a groundbreaking advancement in visual perception, enabling the visualization of hidden objects with significant implications in diverse fields, including autonomous navigation, remote sensing, disaster recovery, and medical diagnostics (Bauer et al., 2015; Lindell et al., 2019a; Scheiner et al., 2020; Laurenzis et al., 2017; Wu et al., 2021; Maeda et al., 2019). By harnessing sophisticated time-of-flight (ToF) configuration, NLOS imaging systems can effectively capture light signals bounced off hidden objects, even when direct line-of-sight visibility is obstructed, as illustrated in Fig. 1(a). The core components of such systems typically include pulse lasers, which emit short bursts of light, and time-resolved detection sensors like Single Photon Avalanche Diode (SPAD) and Time-Correlated Single Photon Counting, which precisely capture the flight of the time that takes for photons to travel from the light source to the hidden object and back to the SPAD. The captured signals, known as transient measurement, undergo reconstruction using various algorithms, including traditional approaches (Velten et al., 2012; Arellano et al., 2017; Liu et al., 2019) and learning-based approaches (Chen et al., 2020; Grau Chopite et al., 2020; Mu et al., 2022; Yu et al., 2023; Li et al., 2023; 2024).

For the traditional approaches, the back projection algorithms (Laurenzis & Velten, 2013; Velten et al., 2012) and the light path transport algorithms (Heide et al., 2019; O'Toole et al., 2018) typically assume isotropically scattering, no inter-reflection, and no occlusions within the hidden scenes. However, these approaches always yield noisy results and lack details. Conversely, the wave propagation approaches (Lindell et al., 2019b; Liu et al., 2020) require no special assumptions and tend to produce better results while being sensitive to scenes with large depth variations. Learning-based approaches (Chen et al., 2020; Mu et al., 2022; Li et al., 2023) leverage the powerful representation capabilities of neural networks, and push NLOS reconstruction to a higher level.

Despite promising results, current NLOS reconstruction algorithms are constrained by the reliance on empirical physical priors and are still confronted with challenges. The primary challenge is Radiometric Intensity Fall-off (RIF), i.e., the intensity of the reflected photons attenuates and the degree of attenuation is related to the surface material of the hidden object. To address this phe-

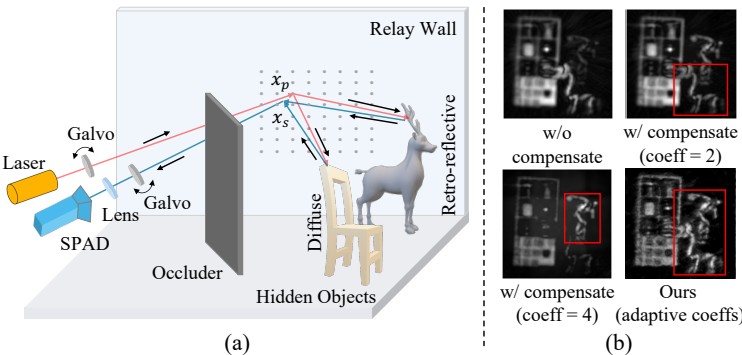

Figure 1: (a) An overview of the NLOS imaging system, including objects with distinct surface materials. (b) Reconstructed images from our method and RSD (Liu et al., 2019) with different compensation coefficients. Near to Far: Dragon, Bookshelf, Statue.

nomenon, quadratic and quartic operations are commonly applied to the light propagation path for retro-reflective and diffuse surfaces, respectively (O'Toole et al., 2018), to compensate for intensity attenuation. However, since various surface materials coexist within the same scene, applying path compensation based on a single material type across the entire scene, as performed in physical priors or previous work, may not effectively counteract the effects of attenuation. Additionally, the problem is exacerbated by the low quantum efficiency of the imaging system, particularly over long distances. As shown in Fig. 1(b), using a single coefficient to compensate for the entire scene can enhance the reconstruction of objects with corresponding material properties, but it will significantly reduce the quality of other objects in the same scene. Another challenge is the limited generalization ability mainly caused by various noises, which consist of the dark count of the SPAD and the noise caused by ambient light (Hernandez et al., 2017). As the data acquisition time decreases, the signal-to-noise ratio (SNR) decreases, resulting in higher noise levels. The Poisson-distributed noise photons degrade the quality of transient measurements especially at low SNR, manifesting as high-frequency aliasing. This phenomenon poses grave challenges to existing approaches, with traditional ones yielding a plethora of artifacts, and learning-based ones experiencing a breakdown in their ability to generalize.

To address the above two challenges, we propose a novel learning-based approach by leveraging the virtual wave phasor field (Liu et al., 2019). Our approach incorporates two key designs: the Learnable Path Compensation (LPC) and the Adaptive Phasor Field (APF). Given that reflected light with different degrees of RIF may be captured simultaneously, the LPC utilizes three physics-based predefined compensation weights to initialize the features of transient measurements for path compensation. Subsequently, a convolutional neural network is trained to implicitly learn and assign optimized compensation coefficients to each scanning point in the transient measurements for the entire scene. By utilizing these optimized coefficients, the LPC adaptively mitigates light wave attenuation in the same scene, as shown in Fig. 1(b), particularly for distant regions. Meanwhile, the APF learns an applicable standard deviation for the Gaussian window of the illumination function, allowing it to dynamically choose the relevant spectrum band for each transient measurement. The emphasis on the effective spectrum enables the discrimination of useful information from noise under distinct SNR conditions.

To demonstrate the efficacy of our proposed approach, we train the approach on a synthetic dataset and subsequently test them on unseen data, including both synthetic and real-world datasets captured from different imaging systems. The exceptional performance on unseen synthetic data and the diverse real-world data highlight the robust generalization capabilities of our approach. Even under challenging conditions, i.e., fast acquisition time and low SNR, our method consistently outperforms competitors. To further increase the diversity of NLOS data, we provide three real-world data captured by our own NLOS imaging system to conduct more comprehensive experiments.

In summary, the contributions of this paper can be listed as follows:

- We propose a novel learning-based approach for NLOS reconstruction, breaking the reliance on empirical physical priors and boosting the generalization capability.

- We design the LPC to adaptively mitigate the light attenuation in the same scene. The embedded learnable physical prior greatly improves the generalization capability across different object materials, especially for long-distance regions.

- We design the APF to prioritize the relevant information from the frequency domain, which improves the generalization capability across transient measurements under distinct SNR conditions.

- Our proposed approach, trained on synthetic data, achieves the best generalization performance on both synthetic and publicly real-world datasets with diverse SNRs. Additional real-world data captured by our own imaging system further showcases the capability of our approach.

## 2 RELATED WORK

### 2.1 TRADITIONAL APPROACHES

In the rapidly advancing field of NLOS imaging, significant progress has been made towards unveiling hidden objects. The groundwork was established by Kirmani et al. (2009), who pioneered the use of time-resolved imaging to navigate photons around obstructions, despite facing computational challenges due to complex multi-path light transport. Efforts to streamline the complex inverse problem have led to the development of back projection approaches, notable for their ability to approximate the geometry of obscured objects through ultrafast time-of-flight information capturing and light geometric relationship (Velten et al., 2012; Arellano et al., 2017). The Light-cone Transform (LCT), marked by the introduction of simple assumptions for light propagation, further facilitated the NLOS reconstruction with unprecedented detail by solving inverse problems in the linear space (O'Toole et al., 2018). The wave propagation approaches like frequency-wavenumber migration (FK) (Lindell et al., 2019b) and Rayleigh Sommerfeld Diffraction (RSD) (Liu et al., 2020; 2019) provided enhanced accuracy for NLOS imaging by considering the interaction between the light wave and multiple hidden object surfaces. Despite considerable progress, traditional algorithms are still limited with challenges in noise effects and complicated scenes.

### 2.2 LEARNING-BASED APPROACHES

Recently, learning-based approaches have been gradually introduced into NLOS imaging. Grau Chopite et al. (2020) proposed the first end-to-end learnable network for NLOS reconstruction. The UNet (Ronneberger et al., 2015) based network regressed the depth from transient measurements directly. However, it is an unstable solution that transforms the non-linear spatial-temporal domain into the linear spatial domain solely by convolution layers. The instability is particularly evident in real-world scenarios, resulting in poor reconstructions. To solve this problem, Chen et al. (2020) developed the physics-based feature propagation module (LFE, Learned Feature Embeddings) to transform different domains, narrowing the domain gap between the synthetic and real-world data. Building on the insights from NeRF (Mildenhall et al., 2021), recent approaches (Mildenhall et al., 2021; Mu et al., 2022) can render the albedo of hidden objects through the radiance field in the unsupervised manner, which consumes large computation time for each inference. Through analysis of transients histogram, Li et al. (2023) produced the first transformer-based framework (NLOST) for capturing local and global correlations, while entailing a substantial computational burden. Yu et al. (2023) introduced a learnable Inverse Kernel (I-K) with attention mechanisms. However, I-K is actually tailored for the point spread function of the imaging system rather than the transient measurements. While the above physics-based approaches (Chen et al., 2020; Li et al., 2023; Yu et al., 2023) consistently improve NLOS reconstruction performance, they still encounter challenges when reconstructing real-world scenes with diverse object materials. Additionally, these approaches overlook the generalization of the real-world transient measurements with low SNRs. In this paper, we present specific solutions tailored to these two challenges.

## 3 METHODOLOGY

### 3.1 IMAGING FORMULATION

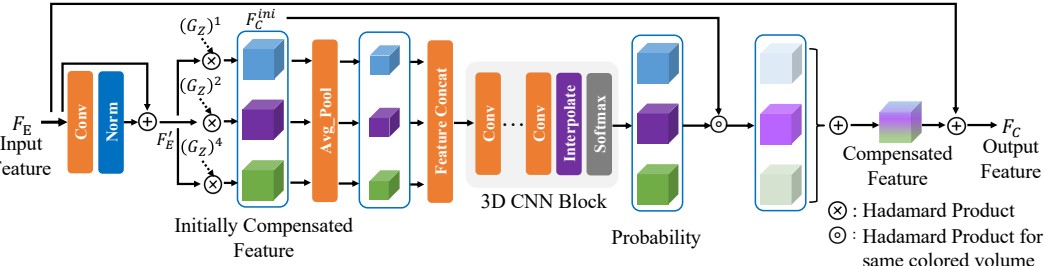

Figure 2: An overview of our proposed approach. Given the transient measurements as input, the approach generates the albedo volume, intensity image, and depth map.

Figure 3: The pipeline of the LPC.

We begin with an impulse response captured from the relay wall, noted as $H(x_p \rightarrow x_s, t)$. With the virtual illumination source wavefront $\mathcal{P}(x_p, t)$, the phasor field at the virtual aperture $\mathcal{P}(x_s, t)$ can be formulated (Liu et al., 2019; 2020) as:

$$\mathcal{P}(x_s, t) = \int_P \mathcal{P}(x_p, t) * \left( \frac{1}{r^z} \cdot H(x_p \rightarrow x_s, t) \right) dx_p, \quad (1)$$

where $*$ denotes the convolution operator, $r$ is the distance between the scanning point and the target point, $x_p$ and $x_s$ represent the illumination point and the scanning point, respectively. The term $1/r^z$ describes the RIF, where $z$ indicates the compensation coefficient associated with different surface materials. Instead of utilizing the empirical compensation coefficient like previous works, we design the LPC to learn this coefficient.

The $\mathcal{P}(x_p, t)$, referred to as the illumination function, is defined as a Gaussian-shaped function modulated with the virtual wave $e^{j\Omega_C t}$, which can be represented as illumination phasor field $\mathcal{P}_{\mathcal{F}}(x_p, \Omega)$ in the Fourier domain as Liu et al. (2019):

$$\mathcal{P}_{\mathcal{F}}(x_p, \Omega) = \delta(x_p - x_{vp}) \cdot \left( 2\pi\delta(\Omega - \Omega_C) * \sigma\sqrt{2\pi} \exp\left( -\frac{\sigma^2\Omega^2}{2} \right) \right), \quad (2)$$

where $\mathcal{F}$ represents the Fourier domain, $x_{vp}$ denotes the position at the virtual light source, $\delta$ is the Dirac function, $\Omega_C$ denotes the central frequency of the wave, and $\sigma$ represents the standard deviation. The standard deviation of a Gaussian is inversely proportional to its pass-band width in the frequency domain, which can be learned and adjusted automatically by our APF module.

The 3D image $I(x_v)$ can be reconstructed from $\mathcal{P}(x_s, t)$ with the wave propagation function $\Phi(\cdot)$, which is modeled by the Rayleigh-Sommerfeld Diffraction integral (Liu et al., 2019):

$$I(x_v) = \Phi\left( \mathcal{P}(x_s, t) \right), \quad (3)$$

where $x_v$ is a point being imaged. Without loss of generality, considering Poisson noise resulting from ambient light and backgro und noise, the computational model of SPAD sensor (Saunders et al., 2019; Grau Chopite et al., 2020) can be written as:

$$H'(x_p \rightarrow x_s, t) \sim \text{Poisson}(H(x_p \rightarrow x_s, t) + B), \quad (4)$$

where $B$ represents detected photons from background noise and dark counts (Bronzi et al., 2015) of SPAD sensors. Poisson($\cdot$) represents the Poisson distribution (Snyder & Miller, 2012).

## 3.2 OVERVIEW

To address the problems mentioned in Section 1, we integrate the proposed LPC and APF modules into the LFE (Chen et al., 2020) framework, which comprises a feature extraction module, a wave propagation module, and a rendering module. An overview of the proposed network is shown in

Fig. 2. Given transient measurements as input, the feature extraction module downsamples the transient measurements in both spatial and temporal dimensions and extracts feature embeddings $F_E$. In addition to albedo, the feature embeddings also encode information such as normals and shapes (Chen et al., 2020; Li et al., 2023)).

Instead of directly applying the wave propagation module to convert features of the transient measurements to the spatial domain, we first employ the LPC to learn different compensation coefficients for each scanning position at the aperture. This allows us to compute the corresponding feature compensation amplitudes, resulting in the compensated feature $F_C$. Subsequently, the APF module predicts the optimal frequency domain window width for the illumination function, which illuminates $F_C$ and generates $F_A$. Finally, the wave propagation module and rendering module convert $F_A$ from the spatial-temporal domain to the spatial domain and render intensity and depth images, respectively.

### 3.3 LEARNING TO COMPENSATE RADIOMETRIC INTENSITY FALL-OFF

To alleviate the aforementioned RIF, we design the LPC module to learn adaptive compensation coefficients and compensate features. An overview of the LPC is shown in Fig. 3. Given the features $F_E$ from the feature extraction module, the LPC first enhances the features using a convolutional layer with normalization, yielding $F'_E$. Let $G_Z$ denote the grid representing the distance from the hidden volume to the relay wall, we predefine three path compensation weights $\{(G_Z)^r, r = 1, 2, 4\}$, which correspond to different attenuation amplitudes of surface materials, as referenced in O'Toole et al. (2018). The weights and enhanced features are multiplied to obtain initially compensated features $F_C^{ini}$, which can be expressed as:

$$F_C^{ini} = \left\{ (G_Z)^1, (G_Z)^2, (G_Z)^4 \right\} \otimes F'_E, \tag{5}$$

where $\otimes$ denotes the Hadamard product.

After that, the initially compensated features are down-sampled across the spatial dimensions using an average pooling layer. Subsequently, the features undergo a series of operations, including convolutional layers, interpolation, and a Softmax operation, which outputs probabilities. To accelerate network convergence, the LPC learns these probabilities instead of directly predicting the compensation coefficients. This design enables the module to explicitly select appropriate compensation weights based on physical constraints. These probabilities are then multiplied by the initially compensated features using the Hadamard product, resulting in the compensated features. Finally, the compensated features are added to the input features, producing the output features $F_C$.

As demonstrated in Section 4.5, our carefully designed LPC module effectively mitigates the RIF issue, enhancing the reconstruction performance for challenging real-world scenes, especially in complex and distant regions.

### 3.4 DENOISING WITH ADAPTIVE PHASOR FIELD

As described in the imaging formulation in Section 3.1, the transient measurement is illuminated by the virtual illumination function. In the frequency domain, the illumination phasor field $\mathcal{P}_\mathcal{F}(x_p, \Omega)$ acts as a Gaussian filter on the feature of transient measurements, modulated to the central frequency $\Omega_C$. It should be noted that not all frequency components contribute positively to the final scene reconstruction, including frequency components associated with noise (Liu et al., 2020; Hernandez et al., 2017). Applying an illumination function to the feature of transient measurements can be understood as a process of selecting a certain effective frequency spectrum band $\triangle\Omega$. The bandwidth of the Gaussian function for the illumination function is decided by the standard deviation, which can be expressed as $\triangle\Omega = 1/(2\pi\sigma)$. The $\triangle\Omega$ is defined as the 3 dB bandwidth. Selecting an appropriate standard deviation is crucial for obtaining clean measurements. However, past works (Liu et al., 2019; 2020) have relied on an empirical standard deviation, which is not conducive to selecting the correct frequency components for the reconstruction of complicated scenarios.

To address this problem, we devise the APF module to adaptively learn the standard deviation, as illustrated in Fig. 4. Given the feature $F_C$, the first step is to transform the feature into the frequency domain along the temporal dimension. Subsequently, the Fourier features are successively convolved across the spatial and spectrum parts to further enhance the features, which may help the module distinguish between useful information and noise more effectively in the frequency domain.

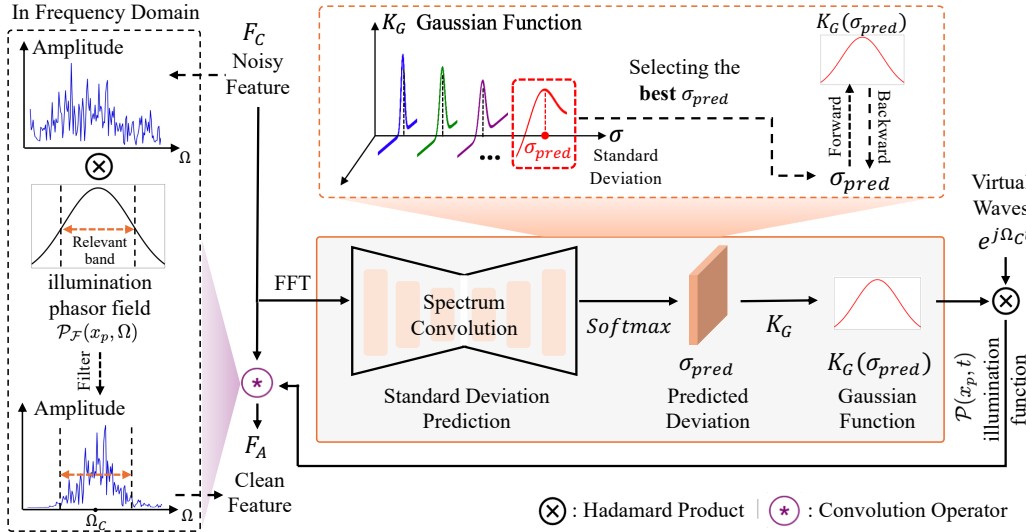

Figure 4: The pipeline of the APF. The module predicts the illumination function with an appropriate bandwidth to compensate for the noisy transient features, outputting clean, denoised features. The left-side illustration is provided solely to explain the filtering principle from a spectral perspective, which is implemented through convolution in the time domain.

We then employ additional fully connected layers to predict the standard deviations $\sigma_{pred}$ from frequency feature representation, generating the adaptive Gaussian function $K_G(\sigma)$. As such, the illumination phasor field can be formulated by the adaptive Gaussian function and the virtual waves $e^{j\Omega_C t}$, as

$$\mathcal{P}_{\mathcal{F}}(x_p, \Omega) = \delta(x_p - x_{vp}) \cdot \left( \mathcal{F}\left(e^{j\Omega_C t}\right) * \mathcal{F}\left(K_G(\sigma_{pred})\right) \right), \quad (6)$$

where

$$\mathcal{F}\left(K_G(\sigma)\right) = \sigma\sqrt{2\pi} \exp\left(-\frac{\sigma^2 \Omega^2}{2}\right). \quad (7)$$

Finally, the input features $F_C$ and the illumination phasor field are convolved across the temporal dimension, as

$$F_A = F_C * \mathcal{P}(x_p, t) = \mathcal{F}^{-1}\left( \mathcal{F}(F_C) \cdot \mathcal{P}_{\mathcal{F}}(x_p, \Omega) \right), \quad (8)$$

where $F_A$ is the output feature at the scanning point, and $*$ means the convolution operator. Notably, Eq. 8 is provided solely to explain APF from a spectral perspective further, which is implemented by the convolution under the time domain.

As demonstrated in Section 4.4 and Section 4.5, the APF module selectively emphasizes useful information and attenuates noise across various SNR conditions within the transient measurements, thereby boosting the generalization capability and improving the reconstruction quality.

### 3.5 Loss Function

The approach is trained in an end-to-end manner. The total loss consists of the intensity loss and the depth loss, balanced by a regularization weight $\lambda$:

$$\mathcal{L} = \mathcal{L}_{\mathcal{I}}(I, \hat{I}) + \lambda\mathcal{L}_{\mathcal{D}}(D, \hat{D}), \quad (9)$$

and

$$\mathcal{L}_{\mathcal{I}}(I, \hat{I}) = \frac{1}{N}\sum_i^N (I_i - \hat{I}_i)^2, \mathcal{L}_{\mathcal{D}}(D, \hat{D}) = \frac{1}{N}\sum_i^N (D_i - \hat{D}_i)^2, \quad (10)$$

where $\hat{I}$ and $I$ denote the reconstructed intensity image and the ground truth, respectively. $\hat{D}$ and $D$ denote the recovered depth map and corresponding ground truth. $N$ denotes the total number of pixels of the intensity image and depth map.

## 4 Experimental Results

Table 1: Quantitative comparisons of different approaches upon the **Seen** test set. The best in **bold**, the second in underline.

| Method | Backbone | Memory | Time | Intensity | | Depth | |
|---|---|---|---|---|---|---|---|
| | | | | PSNR↑ | SSIM↑ | RMSE↓ | MAD↓ |
| LCT (O'Toole et al., 2018) | Physics | 18 GB | 0.11 s | 19.51 | 0.3615 | 0.4886 | 0.4639 |
| FK (Lindell et al., 2019b) | Physics | 26 GB | 0.16 s | 21.69 | 0.6283 | 0.6072 | 0.5801 |
| RSD (Liu et al., 2019) | Physics | 33 GB | 0.23 s | 21.74 | 0.1817 | 0.5677 | 0.5320 |
| LFE (Chen et al., 2020) | CNN | 13 GB | 0.05 s | 23.27 | 0.8118 | 0.1037 | 0.0488 |
| I-K (Yu et al., 2023) | CNN | 14 GB | 0.08 s | 23.44 | 0.8514 | 0.1041 | 0.0476 |
| NLOST (Li et al., 2023) | Transformer | 38 GB | 0.38 s | 23.74 | 0.8398 | 0.0902 | 0.0342 |
| Ours | CNN | 17 GB | 0.24 s | **23.99** | **0.8703** | **0.0874** | **0.0312** |

## 4.1 BASELINES AND DATASETS

**Baseline selection**. To assess the efficacy of our proposed approach, we undertake thorough validations by comparing it against several baseline approaches on the synthetic and real-world datasets. These baselines encompass three traditional approaches commonly used in the field: LCT (O'Toole et al., 2018), FK (Lindell et al., 2019b), and RSD (Liu et al., 2019), as well as three learning-based approaches: LFE (Chen et al., 2020), I-K (Yu et al., 2023), and NLOST (Li et al., 2023).

**Public datasets**. For the synthetic dataset, we utilize a publicly available dataset generated from LFE (Chen et al., 2020). A total of 2704 samples are used for training and 297 samples for testing, denoted as **Seen** test set. Each transient measurement possesses a resolution of $256 \times 256 \times 512$, with a bin width of 33 ps and a scanning area of 2m×2m. To assess the generalization capabilities, we rendered 500 transient measurements from the objects not included in the Seen test set, denoted as **Unseen** test set. For qualitative validation, particularly in complicated scenarios, we employ publicly available real-world data from FK (Lindell et al., 2019b) and also the data from NLOST (Li et al., 2023) with low SNR conditions. For example, instead of the commonly used measurements with 180 minutes acquisition time, we utilize the measurements with 10 minutes acquisition time in FK (Lindell et al., 2019b). We preprocess the real-world data for testing, and the real-world data has a spatial resolution of $256 \times 256$ and a bin width of 32ps.

**Self-captured data**. To further increase the diversity of NLOS data, we also captured additional real-world measurements using our own active confocal imaging system. The system utilizes a 532 nm VisUV-532 laser that generates pulses with an 85 picosecond width and a 20 MHz repetition rate, delivering an average power output of 750 mW. The laser pulses are directed onto the relay wall using a two-axis raster-scanning Galvo mirror (Thorlabs GVS212). Both the directly reflected and diffusely scattered photons are then collected by another two-axis Galvo mirror, which funnels them into a multimode optical fiber. This fiber channels the photons into a SPAD detector (PD-100-CTE-FC) with approximately 45% detection efficiency. The motion of both Galvo mirrors is synchronized and controlled via a National Instruments acquisition device (NI-DAQ USB-6343). The TCSPC (Time Tagger Ultra) records the pixel trigger signals from the DAQ, synchronization signals from the laser, and photon detection signals from the SPAD. The overall system achieves a temporal resolution of around 95 ps. During data acquisition, the illuminated and sampling points remain aligned in the same direction but are intentionally offset to prevent interference from directly reflected photons. As such setting, we capture three transient measurements from customized scenes, each containing different types of surface materials. All measurements were captured over a duration of 10 minutes.

## 4.2 IMPLEMENTATION DETAILS AND METRICS

We implement our approach using the PyTorch framework (Paszke et al., 2019). For optimization, we employ the Adam optimizer (Kingma & Ba, 2014) with a learning rate of $6 \times 10^{-5}$ and a weight decay of 0.95. The $\lambda$ is set to 1. Baseline approaches are implemented using their respective public code repositories. The batch size is uniformly set to 1 for all approaches. Training is conducted for 50 epochs using a single NVIDIA RTX 3090 GPU, except for NLOST, which is trained on Tesla A100 GPUs. Due to memory consumption, NLOST is trained on transient measurements with the shape of $128 \times 128 \times 512$, and the results are interpolated to $256 \times 256$ for comparison.

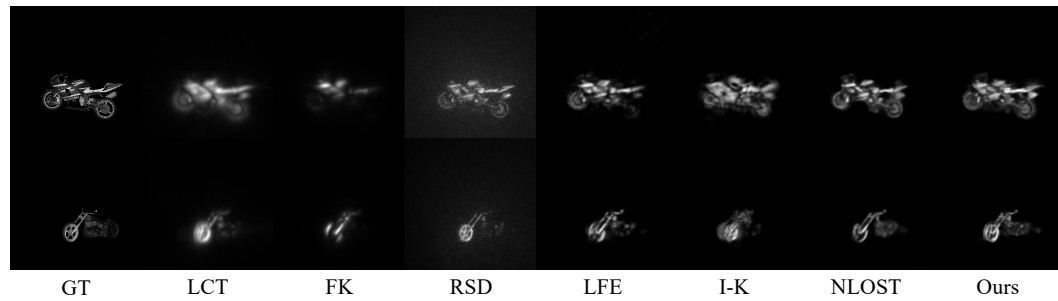

Figure 5: Intensity results recovered by different approaches on the **Seen** test set. GT means ground truth of the intensity images.

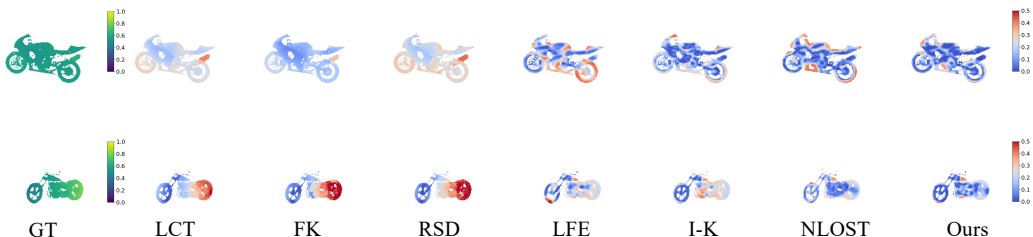

Figure 6: Depth error maps from different approaches on the **Seen** test set. The first column denotes the ground-truth depth map, and the other columns indicate the depth error maps. The color bars show the value of depth and error maps, respectively.

For quantitative evaluation in intensity reconstruction, we adopt peak signal-to-noise ratio (PSNR) and structural similarity metrics (SSIM) averaged on the test set. For depth reconstruction, we compute the root mean square error (RMSE) and mean absolute distance (MAD) for test samples. Following Li et al. (2023), we crop the central region for a more reliable evaluation.

### 4.3 COMPARISON ON SYNTHETIC DATA

**Quantitative evaluation**. The quantitative evaluations presented in Table 1 demonstrate that our approach achieves decent advancements in NLOS reconstruction. For the synthetic results, our approach outperforms all competitors in terms of all evaluation metrics. Specifically, our approach exhibits a substantial enhancement over traditional approaches, achieving a 2.25 dB increase in PSNR compared to the leading approach RSD. Furthermore, when compared with the recent state-of-the-art (SOTA) learning-based approaches I-K and NLOST, our approach still achieves a 0.55 dB and 0.25 dB improvement in PSNR, respectively. The merits of our approach are further substantiated by the highest SSIM for intensity, which underscores the superior capability of our network in preserving the structural integrity of hidden scenes. Additionally, for the depth estimation, our approach reduces the RMSE and MAD metrics by 3.10% and 8.77%, respectively, over the strongest competitor NLOST.

Notably, the existing Transformer-based SOTA approach NLOST requires approximately 38 GB of GPU memory and a substantial amount of inference time. In contrast, our approach achieves higher performance while using only half the memory and requiring less inference time.

**Qualitative evaluation**. We present the qualitative results of intensity images and depth error maps for visualization comparisons, depicted in Fig. 5 and Fig. 6. Regarding the intensity visualization comparisons, LCT reconstructs the main content yet sacrifices details, FK fails to recover most of the structural information, and RSD introduces significant noise in the background. The LFE and I-K perform better than traditional approaches but still lack details. Compared to the SOTA approach NLOST, our approach generates content with greater fidelity and high-frequency details (e.g., the texture of the scene in the first row). In terms of the depth error map, the blue regions dominate the scene in the error map corresponding to our approach, indicating the smallest magnitude of the error. In contrast, traditional approaches as well as LFE demonstrate a greater tendency for errors, as shown by the increased presence of red parts, especially in distant regions (e.g., the right part of

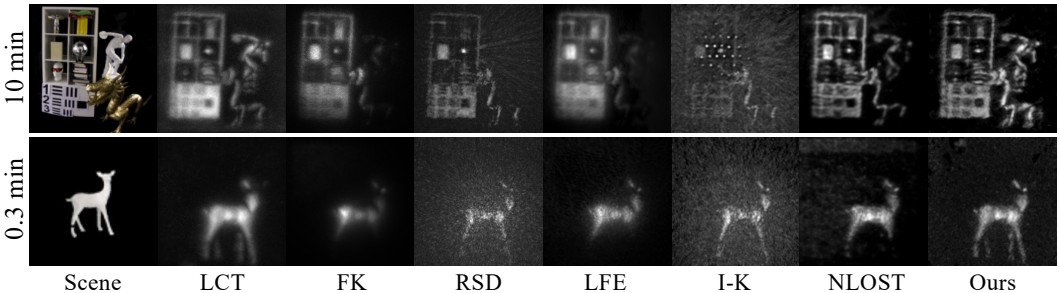

Figure 7: Visualization comparison on the public real-world data (Lindell et al., 2019b; Li et al., 2023). The left annotation indicates the shortest total acquisition time. Zoom in for details.

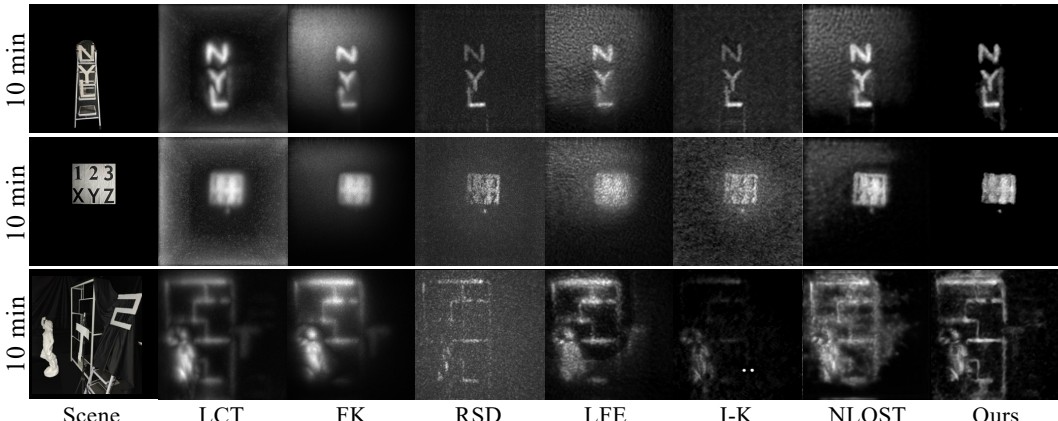

Figure 8: Visualization comparison on our self-captured real-world data. The left annotation indicates the total acquisition time. Zoom in for details.

the motorcycles in the second row). These areas are challenging due to the complex geometrical features and distinct RIF degrees with different kinds of materials. While I-K and NLOST show improvement over the former approaches, they still fail to precisely estimate the depth in the wheel area, where our approach succeeds.

**Generalization evaluation**. To further validate the network's generalization performance, we conduct quantitative tests under varying SNR conditions. Specifically, we test different approaches with the **Unseen** test set under varying SNR levels (10 dB, 5 dB, and 3 dB) of the Poisson noise. Extreme SNR conditions make separating background noise from the limited number of collected photons more challenging, while the new scenes in the Unseen test set validate the performance when transferred to unknown domains. As can be seen in Table 2, in most cases, our approach achieves the best results compared to other approaches. These outstanding results demonstrate the superior generalization performance of our approach in dealing with test data that is distinct from the training data. This superiority is then further verified on various real-world data that has no ground truth below.

## 4.4 COMPARISON ON REAL-WORLD DATA

**Public data**. Results on two public NLOS datasets are presented in Fig. 7. When utilizing measurements with reduced acquisition time, nearly all approaches, except for NLOST and our approach, produce reconstructions with significant noise. The traditional approaches, while reconstructing the main content, produces blurred results. LFE and I-K manage to reconstruct more objects but struggle to capture high-frequency details. NLOST excels in reducing background noise, but it still misses certain details such as the legs of the deer and the intricate patterns of the tablecloth. Our approach shows remarkable resilience to variation in different acquisition times, consistently delivering detailed reconstructions comparable to those of the same objects captured at high acquisition time. The exceptional robustness of our approach demonstrates the superior generalization ability over the existing approaches.

Table 2: Quantitative results on the **Unseen** test set under different SNRs. The best in **bold**, the second in underline.

| Method | Intensity (PSNR↑ / SSIM↑) | | | Depth (RMSE↓ / MAD↓) | | |
|---|---|---|---|---|---|---|
| | 10 dB | 5 dB | 3 dB | 10 dB | 5 dB | 3 dB |
| LCT | 18.92 / 0.1708 | 18.38 / 0.1195 | 18.06 / 0.1007 | 0.6992 / 0.6499 | 0.7490 / 0.1195 | 0.7666 / 0.7197 |
| FK | 21.62 / 0.6496 | 21.62 / 0.6471 | 21.62 / 0.6452 | 0.5813 / 0.5562 | 0.5672 / 0.5427 | 0.5598 / 0.5351 |
| RSD | 22.77 / 0.2045 | 22.48 / 0.1510 | 22.24 / 0.1280 | 0.4198 / 0.3934 | 0.3679 / 0.3358 | 0.3496 / 0.3160 |
| LFE | 23.22 / 0.8122 | 23.15 / 0.7951 | 23.10 / 0.7805 | 0.1036 / 0.0484 | 0.1041 / 0.0491 | 0.1044 / 0.0496 |
| I-K | 23.45 / 0.8386 | 23.38 / 0.8020 | 23.32 / 0.7689 | 0.1045 / 0.0500 | 0.1071 / 0.0571 | 0.1099 / 0.0636 |
| NLOST | 23.63 / 0.7747 | 23.74 / 0.8294 | 23.71 / 0.8135 | 0.0939 / 0.0409 | **0.0909 / 0.0351** | 0.0918 / 0.0368 |
| Ours | **23.91 / 0.8577** | **23.83 / 0.8387** | **23.80 / 0.8645** | **0.0893 / 0.0333** | 0.0914 / 0.0365 | **0.0902 / 0.0332** |

**Self-captured data**. Apart from the public data, we also capture several new scenes with our own NLOS system for further assessment. We present results from three distinct scenes: one depicting retro-reflective letters arranged on a ladder (referred to as 'ladder'), another featuring a panel composed of multiple A4 sheets inscribed with '123XYZ' (referred to as 'resolution'), and the third containing multiple objects with varying surface materials (referred to as 'composite'). As shown in Fig. 8, it can be observed that learning-based approaches still exhibit less reconstruction noise compared to traditional approaches. In the low SNR scenario of the 'ladder', other approaches either fail to reconstruct or produce poor-quality reconstructions. However, our reconstruction exhibits notably high quality, with the ladder legs even discernible. In the heavily attenuated diffuse reflection scenario 'resolution', our approach still manages to reconstruct relatively clear details. In the 'composite' scene, which includes depth variations and multiple surface materials, our approach produces reconstruction with the least noise and the most complete structural information (e.g., the lower edge of the bookshelf and the letter 'S' in the upper right of the scene). The promising outcomes achieved by our approach underscore its superiority over existing approaches.

### 4.5 ABLATION STUDIES

In this section, we ablate the contribution of the modules. As shown in the qualitative results in Fig. 9, the LPC and the APF modules each contribute to improving the performance of the approach in distinct ways, with their combination yielding the best results. Specifically, it can be seen that the network without the proposed modules loses image details and contains significant noise in the reconstruction. In contrast, introducing the LPC module enhances object details (e.g., the deer's legs), and introducing the APF module suppresses background artifacts. When both the APF and the LPC modules are integrated, the network produces images with complete details and clear boundaries.

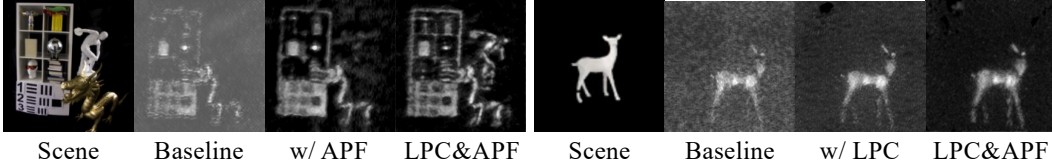

| Scene | Baseline | w/ APF | LPC&APF | Scene | Baseline | w/ LPC | LPC&APF |

Figure 9: Ablation results on public real-world data. The Baseline denotes the LFE with physical prior RSD. The total acquisition time of the left and right scenes is 10 min and 0.3 min, respectively.

### 5 DISCUSSION AND CONCLUSION

In this paper, we propose a novel learning-based approach for NLOS reconstruction including two elaborate designs: learnable path compensation and adaptive phasor field. Experimental results demonstrate that our proposed approach effectively mitigates RIF and improves the generalization capability. Additionally, we contribute three real-world scenes captured by our NLOS imaging system. The future work of our study is twofold. We conduct experiments on the confocal imaging system, with the extension to the non-confocal imaging system being one direction of our future research. The modeling of the SPAD acquisition process still exhibits a certain gap from the real-world sensor, and considering additional factors remains a focus for future research.

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

## A  ADDITIONAL RESULTS

### A.1  MORE REAL-WORLD RESULTS

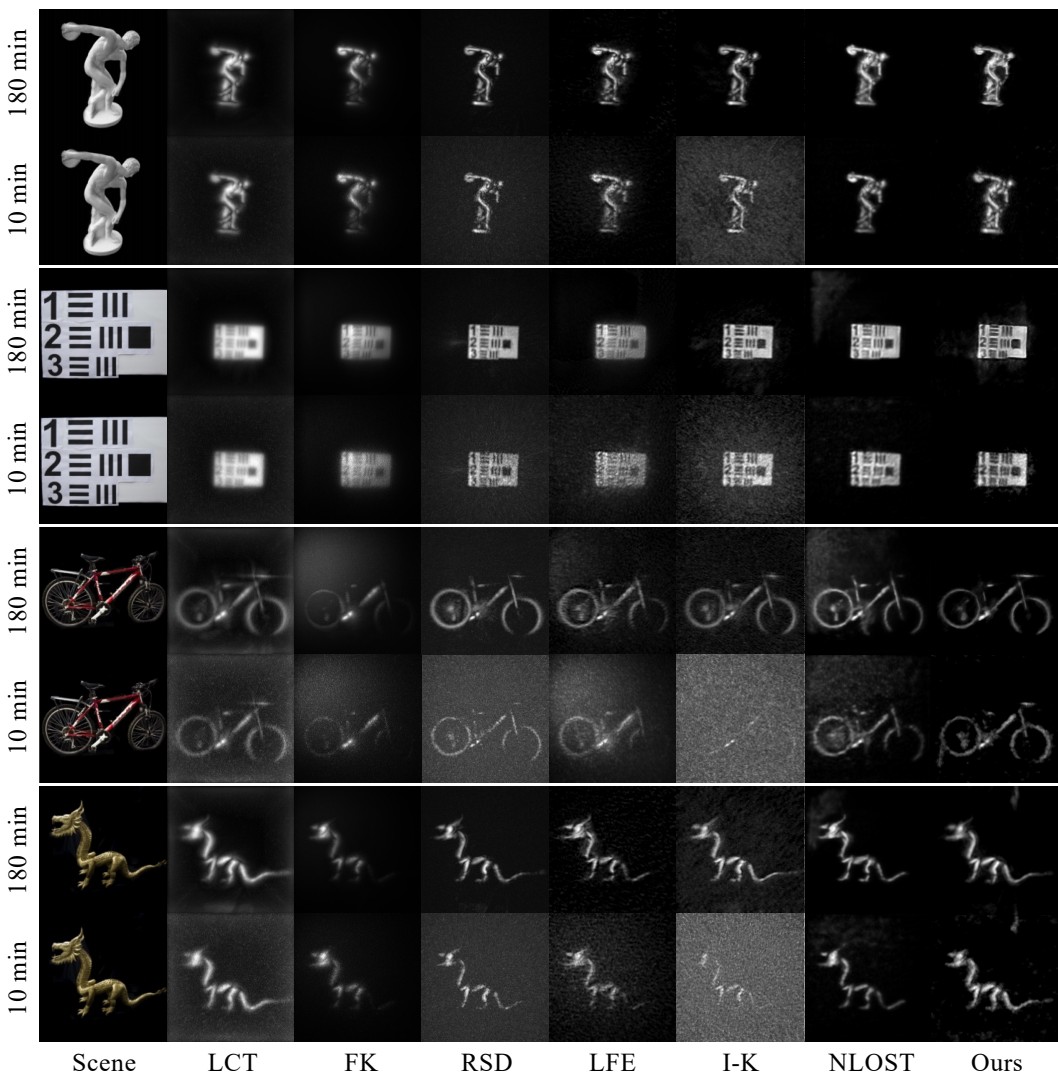

Figure 10: Visualization comparison on the public real-world data (Lindell et al., 2019b). The left annotation indicates the total acquisition time. Zoom in for details.

We provide more real-world results under the public data (Li et al., 2023; Lindell et al., 2019b) and self-captured data. The reconstructed results from different approaches are shown in Fig. 10 and Fig. 11. Compared with other approaches, our approach successfully recovers the clear boundary and full details for each scene, which demonstrates the effectiveness of our proposed method.

### A.2  GENERALIZATION EVALUATION

We provide additional quantitative results to validate the generalization of the approach. In Section 4.3, we present results under different Poisson noise settings. Unlike those settings, in this section, we vary the photon acquisition efficiency to affect the SNR of the transient measurements. With the same level of background noise, lower photon acquisition efficiency results in fewer collected photons and a lower SNR. As can be seen in Table 3, in most cases, our approach achieves the best results compared to other approaches. This demonstrates that our network effectively compensates for photon acquisition and reduces noise, resulting in more robust and clearer reconstructed images.

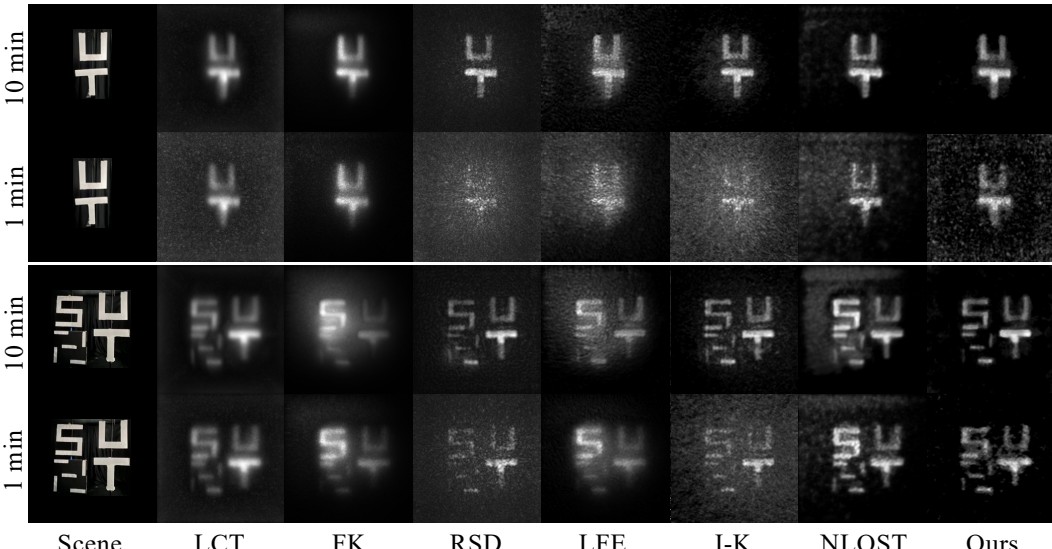

Figure 11: Visualization comparison on the self-captured real-world data. The left annotation indicates the total acquisition time. Zoom in for details.

Table 3: Quantitative results on the **Unseen** test set under different SNRs. The SNR is controlled by adjusting the photon acquisition efficiency. The best in **bold**, the second in underline.

| Method | Intensity (PSNR↑ / SSIM↑) | | | Depth (RMSE↓ / MAD↓) | | |
|---|---|---|---|---|---|---|
| | 48 dB | 28 dB | 25 dB | 48 dB | 28 dB | 25 dB |
| LCT | 17.47 / 0.1863 | 14.44 / 0.0452 | 11.54 / 0.0287 | 0.5581 / 0.5166 | 0.8143 / 0.8024 | 0.8207 / 0.8109 |
| FK | 20.30 / 0.5981 | 15.87 / 0.0693 | 13.96 / 0.0305 | 0.7861 / 0.7847 | 0.8453 / 0.8360 | 0.8178 / 0.8065 |
| RSD | 20.78 / 0.2286 | 14.24 / 0.0320 | 11.72 / 0.0219 | 0.4415 / 0.4042 | 0.3972 / 0.3609 | 0.3567 / 0.3228 |
| LFE | 21.25 / 0.7154 | 14.84 / 0.0670 | 12.75 / 0.0353 | 0.0202 / 0.0158 | 0.3176 / 0.3075 | 0.3515 / 0.3398 |
| I-K | 21.05 / **0.7759** | 15.19 / 0.0559 | 12.54 / 0.0302 | 0.0246 / 0.0177 | 0.2502 / 0.2380 | 0.3686 / 0.3545 |
| NLOST | 21.19 / 0.7386 | 17.84 / 0.1387 | 15.36 / 0.0647 | **0.0096** / 0.0076 | 0.1984 / 0.1871 | 0.2477 / 0.2385 |
| Ours | **21.35** / 0.7654 | **19.06** / **0.4711** | **17.86** / **0.4078** | 0.0097 / **0.0075** | **0.1096** / **0.1012** | **0.1480** / **0.1396** |

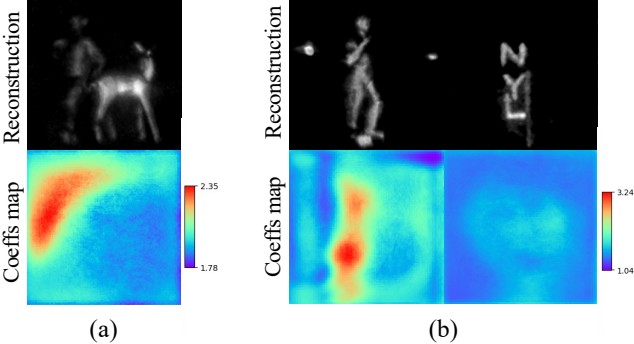

(a)          (b)

Figure 12: Visualization of the predicted compensation coefficients by LPC. 'Reconstruction' denotes the results of our method, while the 'Coeffs map' denotes the predicted compensation coefficients for the corresponding scenes.

## A.3 VISUALIZATION OF COMPENSATION COEFFICIENTS

In this section, we provide visualization results of the predicted compensation coefficients from LPC. It can be observed from Fig. 12(a) that LPC assigns larger compensation coefficients to the object farther away (i.e., the statue on the left) in the scene, compared to the closer object (i.e., the deer on the right). For single-object scenes from Fig. 12(b), LPC assigns compensation coefficients corresponding to the material properties of the objects (with the 'man' scene containing diffuse

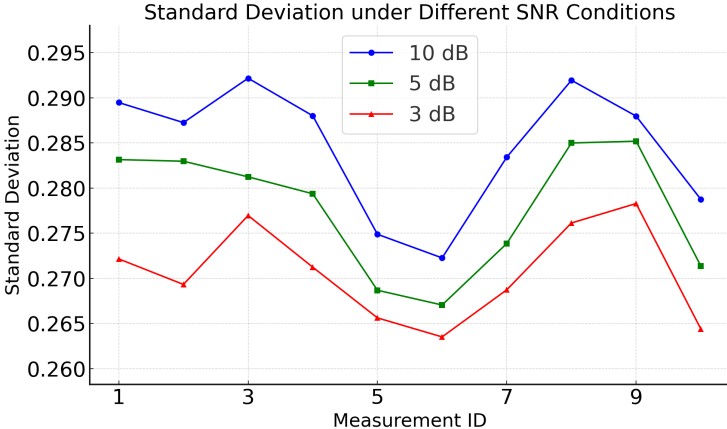

Figure 13: Visualization of the standard deviation predicted by APF. Different colored lines in the figure illustrate how the predicted standard deviation varies under different SNR conditions.

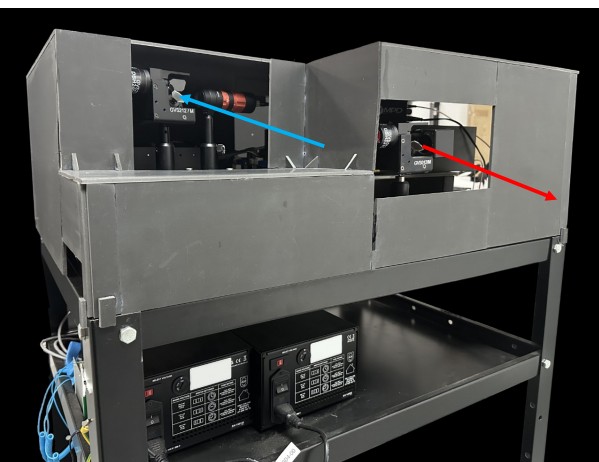

Figure 14: Our own NLOS imaging system.

material, and the 'ladder' scene containing retro-reflective material) in the areas where the objects are placed. Since an end-to-end network tends to learn and converge with the goal of optimizing the final output (reconstructed image and depth), the intermediate results in visualizations may not perfectly align with the theoretical values (quadratic or quartic). However, we can observe that the compensation coefficients indeed vary according to different materials.

### A.4 VISUALIZATION OF STANDARD DEVIATIONS

In this section, we provide visualization results of the standard deviation from APF. We randomly select 10 transient measurements from the Seen test set. A lower standard deviation of the Gaussian window in the frequency domain typically indicates a narrower frequency response, leading to stronger high-frequency noise suppression. As shown in Fig. 13, the standard deviation exhibits a decreasing trend as the SNR decreases, which means APF predict a tighter Gaussian window for features to filter out more noise.

## B    IMAGING SETUP

**System details**. Details of our system are shown in Fig. 14.

**Capturing details**. The materials in the self-captured scenes can be divided into two types: retro-reflective type (e.g., a bookshelf, and the letters 'NYLT' covered with reflective tape) and diffuse

type (e.g., the number '2' covered with white printer paper, a cardboard printed with '123XYZ', and the plaster statues).

## C    DETAILS OF THE NETWORK STRUCTURE

**Feature extraction module**. Given transient measurements as input, the feature extraction module is responsible for downsampling and extracting feature vectors, as well as enhancing the transient measurements. As described in LFE (Chen et al., 2020), the module consists of two branches. The first branch contains a convolutional layer (kernel size = 3, stride = 1), while the second branch contains a ResNet block (He et al., 2016). Each ResNet block includes two convolutional layers and one LeakyReLU layer. The output feature dimension of each branch is 1, and the spatial size of the output is $4\times$ smaller than the input data. The two branches are concatenated along the feature dimension and then output.

**Spectrum convolution**. Firstly, the features of the transient measurements are transformed into the frequency domain using a Fourier transform. Subsequently, a series of 3D convolutional layers with a stride of (1, 2, 2) is applied to reduce the spatial dimensions of the feature vectors, with each convolutional layer followed by a ReLU activation layer. The extracted features are then processed using a 1D convolutional layer, normalized with LayerNorm, and passed through a fully connected layer to fuse the spectrum dimension into a scalar value. Finally, a sigmoid activation computes the desired standard deviation.

**Wave propagation module**. We utilize the physics-based approach RSD (Liu et al., 2019; 2020) as a wave propagation module. The module transforms the features from the spatial-temporal domain into the linear spatial domain.

**Rendering module**. The rendering network consists of one convolutional layer and two custom convolutional blocks. Each convolutional block includes one convolutional layer and two ResNet blocks. The structure of the ResNet blocks is the same as in the feature extraction module. The network takes the output from the wave propagation module as input and applies the first convolutional block to obtain enhanced features. Next, the enhanced features are concatenated with the input features along the spatial dimension and processed by the second convolutional block. To ensure more stable training, we employ a residual structure, where the output of each convolutional block is added to the input features, producing the final output of the rendering module (i.e., intensity and depth images).

