# OpenReview forum: "Generalizable Non-Line-of-Sight Imaging with Learnable Physical Priors"
_ICLR.cc/2025/Conference — Submitted to ICLR 2025_

### Official Review · Reviewer_Eqe9 · 2024-10-21

**Soundness:** 3
**Presentation:** 2
**Contribution:** 2
**Rating:** 5
**Confidence:** 2

**Summary:**

The paper presents a new end-to-end network architecture for reconstructing non-line-of-sight (NLOS) scenes from low signal-to-noise ratio (SNR) single-photon avalanche diode measurements. The network architecture is inspired by the virtual wave phasor field, and introduces two new modules, ie the learnable path compensation module and the adaptive phasor field module.
The proposed method is shown to perform better than previous NLOS approaches (both learned and physics-based) on a public dataset and a new dataset introduced in the paper.

**Strengths:**

- the proposed architecture incorporates insights from virtual wave phasor fields.
- the method achieves state-of-the-art performance for NLOS reconstruction.

**Weaknesses:**

- I found the presentation of the method hard to follow, especially for someone who is not an expert in virtual phasor fields, and in particular the paper "Phasor field diffraction based reconstruction for fast non-line-of-sight imaging systems". I believe that an ICLR submission should be accessible to a larger audience.
    - It was unclear to me how equation (4) and (1) are related, since $H$ is inside an integral in (1)
    - the notation is confusing in equation 3: how are the coordinates $x,y$ in $I$ related to $x_s$? Is $x_s$ referring to a 2D coordinate whereas $x$ and $y$ are 1-D coordinates respectively?
    - there should be at least some intuitive explanation of the function $\Phi$ in equation 3.

- Despite the state-of-the-art performance, the gap with the previous best performing model is relatively mild (often less than 0.5 dB).

**Questions:**

- why is the rendering model called 'rendering'? Is not clear to me why the relatively generic 2D CNN architecture should be a renderer.

**Details Of Ethics Concerns:**

I believe there should be at least some discussion of the positive negative usages of this method, especially in terms of military applications.

---

> ### Author Response · Authors · 2024-11-22
> **Official Comment by Authors**
>
> Thank you for your valuable time and comments. The main concerns are addressed below.
>
> **Q1: The relation between the Eq. (1) and Eq. (4).**
>
> * In Eq. (1), H represents the transient measurement under ideal conditions, whereas H' in Eq. (4) incorporates noise (as claimed in lines 205--206 of the manuscript). During network training and testing, H in Eq. (1) is replaced by H' to better reflect real-world conditions. We hope this explanation clarifies your concerns.
>
> **Q2: Explanations of the notations in Eq. (3).**
>
> * Thanks for your reminder. The coordinates in $I$ are intended to represent point coordinates in the 3D image, which have three dimensions. However, upon review, the coordinates notation in Eq. (3) is indeed imprecise, which should be revised to $x_v=(x,y,z)$ in alignment with [1,2]. $I(x_v)$ represents the 3D image, reflecting the shape of the hidden object. The relationship between $x_s$ and $x_v$ is described by the RSD integral. For details and derivations of the RSD integral, please refer to [1].
>
> **Q3: Intuitive explanation of the function $\Phi$ in equation 3.**
>
> * Thanks for your suggestion. Since we use RSD as the physical prior for our method, the function $\Phi$ here can be represented as the RSD integral. Based on the explanation of the function by Liu et al. [1,2], the RSD operator is used to model the propagation in intensity-modulated light scenes. For more details, please refer to [1].
>
> **Q4: Synthetic performance concern.**
>
> * Compared to previous CNN-based methods, our approach achieves a notable improvement of at least 0.55 dB, which is significant relative to the marginal improvements typically observed among other CNN-based models. Additionally, our method requires only half the memory consumption and achieves faster inference compared to the transformer-based method NLOST. This balance between performance and resource efficiency highlights the advantages of our method, making it a potential solution for real-world applications.
>
> **Q5: Regarding the naming of the rendering module.**
>
> * Our approach is designed based on the LFE framework, as claimed in lines 214--215. The output of the wave propagation module is a three-dimensional feature volume, which is then rendered by the Rendering Net to produce the 3D albedo. This rendering process can be understood as a decoding procedure for the feature information. Thanks for your comments, we will include additional explanations in the revised paper.
>
> Reference:
>
> [1] X. Liu et al., “Non-line-of-sight imaging using phasor-field virtual wave optics,” Nature, vol. 572, no. 7771, pp. 620–623, Aug. 2019.
>
> [2] X. Liu, S. Bauer, and A. Velten, “Phasor field diffraction based reconstruction for fast non-line-of-sight imaging systems,” Nat. Commun., vol. 11, no. 1, p. 1645, Apr. 2020.

---

> > ### Comment · Reviewer_Eqe9 · 2024-11-27
> >
> > Many thanks for your answers to the questions.
> >
> > From the answer, I believe that there was no misunderstanding regarding my original evaluation of the work (except for the meaning of $H$ and $H'$, which now is clear), and I thus prefer to keep my original score.

---

> > > ### Author Response · Authors · 2024-12-02
> > > **Official Comment by Authors**
> > >
> > > Dear reviewer Eqe9,
> > >
> > > Thank you for your response. We are pleased to hear that our explanation has resolved your confusion. Should you have any remaining concerns on our work, please kindly let us know and we are happy to address them in the ending period of dicussion.
> > >
> > > Otherwise, if all aspects are now clear and you see no major weaknesses in our carefully revised manuscript, we would like to kindly ask you to consider updating your score, which could reflect the consensus that has been arrived during the discussion.
> > >
> > > Thank you once again for your valuable feedback and time.
> > >
> > > Sincerely,
> > >
> > > The Authors

---

### Official Review · Reviewer_iqtw · 2024-11-03

**Soundness:** 2
**Presentation:** 2
**Contribution:** 2
**Rating:** 5
**Confidence:** 4

**Summary:**

This paper aims at solving the problem that existing NLOS methods rely on some empirical physical prior during their implement. To address this problem, the authors introduce a Learnable Path Compensation (LPC) module and an Adaptive Phasor Field (APF) module to optimize 3 distance-related weights and a Gaussian window separately. The results on both synthetic data and real data demonstrate the effectiveness of the proposed method.

**Strengths:**

The originality of this paper arises from the improvement of an existing NLOS reconstruction network LFE (Chen et al., 2020). The proposed method can achieve global intensity consistency and avoid manual adjustment of empirical parameters.

**Weaknesses:**

IMO, the physical explanation of this paper is inaccurate and misleading, especially the explanation of the learning of the Gaussian window. And the improvement of the results is limited quantitatively and qualitatively, especially compared with NLOST.

**Questions:**

1.	$P(x_p,t)$ is the virtual illumination function in the Phasor-field methods (Liu et al.,2019), but it is not necessary a Gaussian-shaped function. It can be any type of the light source function, such as ambient light. Please refer to Table S.2 of Liu’s paper. Therefore, the physical explanation of learning a Gaussian window is not correct in my opinion.
2.	Even though some light source functions are Gaussian-shaped, such as pulsed point light function, the frequency spectrum band should be a certain value. It shouldn’t be a changeable value for different scenarios. So, from this point of view, the explanation of learning a Gaussian window is also not correct.
3.	The $I(x,y)$ in Eq.(3) should be the image of the virtual imaging system, it is not the hidden object as claimed in the paper, please refer to the explanation of Eq.(6) in Liu’s paper.
4.	Could you explain in what dimensions of measurements you did the FFT to get the spectrum of the light source function? I think you can only get the spatial and temporal sampling frequency if you operate FFT on raw ToF measurement, rather than illumination spectrum.
5.	Lines 78-79 claims that the proposed method can solve the dark counts noise of SPAD and ambient light. But I didn’t find the detailed solution in the main paper. There is just a simple description of the definition in Eq.(4).
6.	Why did you choose 3 path compensation weights? I didn’t find the related content in O’Toole et al., 2018. Could you clarify it for me?
7.	Which method is chosen as your baseline in Fig.9. I suppose to be LFE according to Fig.2. But the baseline results shown in Fig.9 are different with the LFE results in Fig.7. So, it’s hard to demonstrate the effectiveness of this ablations studies.

## Reference
- Chen et al, Learned feature embeddings for non-line-of-sight imaging and recognition, TOG 2020
- Liu et al, Non-line-of-sight imaging using phasor-field virtual wave optics, Nature, 2019
- O’Toole et al, Confocal non-line-of-sight imaging based on the light-cone transform, Nature 2018

------
**post-rebuttal:**

My questions are properly answered. However, it also reveals the authors' original motivation (the physical intuition and explanation) is somewhat far-fetched. As a result, I raise my score to borderline reject.

---

> ### Author Response · Authors · 2024-11-22
> **Official Comment by Authors**
>
> Thank you for your valuable time and comments. The main concerns are addressed below.
>
> **Q1: Selection of Gaussian-shaped function as light source function is inaccurate.**
>
> * There might be some misunderstanding here. We do NOT claim that the illumination function must be a Gaussian-shaped function. In this paper, we aim at **active NLOS imaging based on a SPAD      camera** and thus we select the commonly used Gaussian-shaped function as the illumination function. This is also consistent with Table S.2 of Liu’s paper [2], where different illumination functions are coupled with different imaging systems.
>
> **Q2: Learning a Gaussian window with a changeable spectrum band is inaccurate.**
>
> * We respectfully cannot agree. As highlighted in the concurrent work [6], the illumination function with the Gaussian shape works as a band-pass filter in the frequency domain, and only a certain range of the spectrum is necessary for recovering hidden objects. This is highly consistent with our motivation, as discussed in lines 255--260 of our manuscript.
>
>   Since different scenes exhibit varying spectrum under different noise conditions, we introduce the APF to learn the standard deviation for relative band selection and adaptive denoising.
>
>   Additionally, we visualize the output of the APF at:
>
>   [Anonymous Link]https://picx.zhimg.com/80/v2-15039835e627e26b62ef2ceb7f5f0e92.png
>
>   which includes evaluation with 10 transient measurements in the Seen test set under distinct SNR conditions. The results show that the standard deviation decreases as the SNR decreases, indicating that APF learns an adaptively Gaussian window and applies stronger high-frequency noise suppression under lower SNR conditions.
>
> **Q3: Explanation of the $I(x, y)$.**
>
> * We sincerely thank the reviewer for pointing out the imprecise description regarding $I(x_v)$. $I(x_v)$ should represent the 3D image of the hidden scene. The intended meaning of lines 202--203 in the manuscript is that the wave propagation function propagates $x_s$ to points in $I(x_v)$, which reflect the shape of the hidden objects (point coordinates denoted as $x_v$, consistent with [2,3]). Following the reviewer's suggestion, we will make the following revisions to ensure the theoretical accuracy of the manuscript:
>
>   * Modify Eq. (3) to $I(x_v)= \Phi \left ( \mathcal{P}(x_s, t) \right )$,
>   * Revise lines 202–-203 to redefine $I(x_v)$.
>
>   These changes will not affect the experimental conclusions in the manuscript.
>
> **Q4: Question of the FFT for light source function.**
>
> * Firstly, Eq. (2) defines the frequency-domain representation of the illumination function, illustrating that the illumination phasor field, while Gaussian-shaped, performs a filtering operation centered at $\Omega_C$. Secondly, we do not perform an FFT on the illumination function itself. The purpose is to enable the network to learn data characteristics in the frequency domain and predict the optimal standard deviation.
>
> **Q5: Concerns for the lines 78--79.**
>
> * Hernandez et al. model the noise inherent in NLOS imaging [4], accounting for the effects of ambient light and dark counts. Subsequently, Chen et al. synthesize datasets based on this mathematical model for training the LFE algorithm [5]. To maintain consistency with prior work, we continue using the dataset simulation tools employed by [5], which is why we refer to ambient light and dark counts as the noise sources of interest in our paper. The motivation of our paper is not to model noise but to discuss how to improve the approach's generalization performance from the perspective of noise present in the measurements. Section 4.5 validates the proposed approach's ability to suppress noise effectively.
>
> **Q6: Reasons of choosing 3 path compensation weights.**
>
> * For quadratic and quartic compensations, please refer to the description of "Validating radiometric falloff" in the supplementary materials of [1], Page 2. Additionally, to enhance the stability of network training, we include an extra linear compensation. Thanks for pointing this out, we will include additional explanations in the revised paper.
>
> **Q7: The LFE results in Fig.7 is different from the LFE (baseline) in Fig.9.**
>
> * In Fig. 9 of the manuscript, the LFE (baseline) is configured with the same physical prior (RSD) as our proposed method, which helps eliminate the influence of the physical prior on the conclusions of the ablation. In contrast, the LFE shown in Fig. 7 employs FK as the physical prior, consistent with the official codebase. Thanks for pointing this out, we will include additional explanations in the revised paper.

---

> ### Author Response · Authors · 2024-11-22
> **Official Comment by Authors**
>
> **Q8: The limited improvement compared with NLOST.**
>
> * It needs to be noted that NLOST is based on the transformer architecture and the resolution of the input measurement is limited due to computational resource restriction. Relying on a more lightweight CNN structure, our approach achieves improved results but only requires **half the memory consumption and less inference time compared to NLOST**, which greatly faciliates the reconstruction of higher-resolution scenes. This balance between performance and efficiency highlights the advantages of our approach.
>
>
> Reference:
>
> [1] M. O’Toole, D. B. Lindell, and G. Wetzstein, “Confocal non-line-of-sight imaging based on the light-cone transform,” Nature, vol. 555, no. 7696, pp. 338–341, Mar. 2018.
>
> [2] X. Liu et al., “Non-line-of-sight imaging using phasor-field virtual wave optics,” Nature, vol. 572, no. 7771, pp. 620–623, Aug. 2019.
>
> [3] X. Liu, S. Bauer, and A. Velten, “Phasor field diffraction based reconstruction for fast non-line-of-sight imaging systems,” Nat. Commun., vol. 11, no. 1, p. 1645, Apr. 2020.
>
> [4] Q. Hernandez, D. Gutierrez, and A. Jarabo, “A Computational Model of a Single-Photon Avalanche Diode Sensor for Transient Imaging,” Feb. 23, 2017, arXiv: arXiv:1703.02635.
>
> [5] W. Chen, F. Wei, K. N. Kutulakos, S. Rusinkiewicz, and F. Heide, “Learned feature embeddings for non-line-of-sight imaging and recognition,” ACM Trans. Graph., vol. 39, no. 6, pp. 1–18, Dec. 2020.
>
> [6] I. Cho, H. Shim, and S. J. Kim, “Learning to Enhance Aperture Phasor Field for Non-Line-of-Sight Imaging,” Jul. 28, 2024, arXiv: arXiv:2407.18574.

---

> > ### Comment · Reviewer_iqtw · 2024-11-25
> > **Thanks for the response**
> >
> > Q1: I believe the authors may have misunderstood my initial question. It is indeed correct that a commonly used Gaussian-shaped function can be chosen as the illumination function. However, in the original Phasor-field method, the illumination function is not equivalent to the actual illumination source used in active NLOS scenarios. This is precisely why it is referred to as a "virtual illumination function." For instance, Liu et al. demonstrated three types of virtual illumination functions, including Gaussian-shaped functions, in their paper.
> >
> > The issue lies in how you attempt to explain the physical reasoning behind your proposed method by partially referencing the theory of the Phasor-field method. In doing so, you seem to have conflated several key concepts, such as treating the virtual illumination function as the actual light source, and misinterpreting 𝐼(𝑥,𝑦), as pointed out in my previous Question 3. While these misunderstandings do not affect the experimental conclusions, they do highlight weaknesses in the physical explanation provided.
> >
> > Q2: Thank you for clarifying that the learnable Gaussian window effectively reduces noise. However, the left part of Fig. 4 is somewhat misleading, as it might give readers the impression that applying FFT to the illumination function is an integral part of the pipeline.

---

> > > ### Author Response · Authors · 2024-11-26
> > > **Official Comment by Authors**
> > >
> > > **For Q1:**
> > >
> > > * We sincerely appreciate the reviewer’s further feedback, which greatly helps us deepen our understanding of the physical concepts underlying the RSD approach and eliminate any misunderstandings of the initial questions. We would like to take this opportunity to further clarify our motivation.
> > >
> > >   In this paper, we draw inspiration from RSD but aim to tackle the task from another perspective. Specifically, we start from observing a commonly used illumination function and improve it based on the Gaussian operation mechanism. In other words, we build upon and expand one of the theoretical branches of RSD, as also demonstrated in another concurrent work [1]. This highlights the strength of RSD and underscores its potential for further exploration.
> > >
> > >   Following your constructive suggestion in the initial questions, we have immediately revised the theory description and our motivations which may result in misunderstanding in the manuscript. Once again, we thank you for carefully reviewing the paper and patiently pointing out the theoretical shortcomings, which has significantly enhanced our manuscript.
> > >
> > >   [1] Learning to Enhance Aperture Phasor Field for Non-Line-of-Sight Imaging, ECCV 2024.
> > >
> > >
> > > **For Q2:**
> > >
> > > * Thank you for your insightful comment. The left-side illustration is meant to provide an intuitive explanation of how the Gaussian function works as a band-pass filter in the frequency spectrum. In the actual computation progress, we just adopt convolution in the time domain. To eliminate any potential misleading, we now include clarification in both the main text and the caption of Fig. 4 in the revised manuscript.
> > >
> > > **In summary, as pointed out by the reviewer, the original inaccurate descriptions “do not affect the experimental conclusions”, and we have carefully revised them accordingly. We hope the current manuscript could be better recognized by integrating valuable suggestions from all expert reviewers, and thus have the opportunity to serve as a more solid solution in this important field. We believe this kind of opportunity is just the key value of open discussion in the venue like ICLR.**

---

> > > ### Author Response · Authors · 2024-12-01
> > > **Official Comment by Authors**
> > >
> > > Dear Reviewer iqtw,
> > >
> > > As the discussion period is ending soon, we would like to send a kind reminder about our latest responses and the revised manuscript. We have addressed each of your concerns in detail and incorporated your suggestions into our revisions. If there are still remaining concerns, we will do our best to provide clarifications as soon as possible. Otherwise, we look forward to your final justification.
> > >
> > > Once again, we appreciate your time and consideration.
> > >
> > > Sincerely,
> > >
> > > The Authors

---

> ### Author Response · Authors · 2024-11-28
> **Official Comment by Authors**
>
> Dear reviewer iqtw,
>
> We sincerely thank you for your valuable time and suggestions. As today is the final day for manuscript revisions, we aim to improve the manuscript as much as possible before the deadline. Should you have any further questions, we are happy to provide additional clarification and resolve them.
>
> We look forward to your positive feedback if our previous responses and revised manuscript adequately address your major concerns.
>
> Thanks once again.
>
> Sincerely,
>
> The Authors

---

### Official Review · Reviewer_M5tk · 2024-11-03

**Soundness:** 4
**Presentation:** 2
**Contribution:** 3
**Rating:** 6
**Confidence:** 4

**Summary:**

This paper proposes a learning-based Non-line-of-sight (NLOS) reconstruction method based on the standard virtual wave phasor field formulation that learns radiometric intensity falloff coefficients to compensate for falloff due to varying materials present in the scene and frequency cutoff values to compensate for varying SNR conditions.

EDIT: upgraded score from a 5 to a 6 given authors response.

**Strengths:**

The paper is well motivated and distinguished clearly from previous works, as the method is structured around leveraging flexibility in the virtual phasor field formulation that was not used by prior methods to improve NLOS reconstruction performance. The improvement in reconstruction performance is clearly demonstrated with extensive experiments on existing and newly gathered datasets across a large number of NLOS reconstruction methods.

**Weaknesses:**

Though the method shows strong performance compared to baselines, I'm not sure the method is better for the reasons that the authors say, rather than just because the method has more parameters and is trained on more data than previous methods. In particular, the authors make several claims about why their method works better that I am not sure are precisely substantiated in the paper.

First, the authors claim that the using adaptive coefficients for the radiometric intensity falloff is better because different materials will have different coefficients and their design can adapt to the varying materials. For example, diffuse objects should have a falloff coefficient of 4 and retro-reflective 2. While the authors do visualize in Figure 1 that their method using adaptive coefficients can recover both diffuse and retroreflective objects simultaneously, I could not find in the paper any visualizations of the compensation coefficients. I would recommend the authors include a image plot of the compensation coefficients, and would expect that regions per object would have their own coefficient each.

Second, the authors claim that the adaptive phasor field improves performance across a variety of SNR conditions. However, in Table 2, which contains the quantitative results across a variety of SNR conditions, the decrease in performance as SNR decreases for the author's proposed method is similar to the other methods. Thus, given the data presented in the paper it seems that it is not the case the author's proposed method is more robust in lower SNR conditions than other methods, despite having uniformly better performance across SNR settings. I'm not sure if it's because the SNR conditions in the testing sets are not varied enough, or if the adaptive phasor field doesn't actually help in diverse SNR settings. Regardless, I recommend that the authors provide some visualization of the standard deviations for the gaussian of the illumination phasor field predicted by their adaptive phasor field to better help the reader understand what the method is actually doing.

**Questions:**

Major:
Reiterating some points from weaknesses:
- I would recommend the authors include a image plot of the compensation coefficients predicted by the learnable path compression, and would expect that regions per object would have their own coefficient each, because I wonder if the model can actually infer that different materials should have different compensation coefficients like the authors claim.
- I recommend that the authors provide some visualization of the standard deviations for the gaussian of the illumination phasor field predicted by their adaptive phasor field to better help the reader understand what the method is actually doing, because I have questions about if the adaptive standard deviations used in the wave propagation are actually reducing the noise in low SNR environments like the authors claim. Specifically, I would expect for low SNR environments the standard deviation to be lower and vice versa, because low SNR implies higher noise and thus the model should use a tighter gaussian during wave propagation to filter out more noise.

I would be willing to upgrade my score if the authors could provide more evidence that the method works the way they claim.

Minor:
- I don't quite understand in equation (8) why the inverse Fourier transform is taken, because according to Figure 2, $F_A$ goes through a Fourier transform before getting input into the wave propagation model. It seems to me to be more concise and efficient to keep the result of convolving the illumination phasor field with the compensated features in the frequency domain for wave propagation rather than transforming it back to the spatial/time domain.
- For the real data at varying SNR levels, 10 dB, 5 dB, and 3 dB, how did the authors capture this data? I would expect that the exposure time is different between captures, but I couldn't see any details about this in the paper.

---

> ### Author Response · Authors · 2024-11-22
> **Official Comment by Authors**
>
> Thank you for your valuable time and comments. The main concerns are addressed below.
>
> **Major:**
>
> **Q1: Visualization of the compensation coefficients from LPC module.**
>
> * We provide visualizations of the compensation coefficients predicted by LPC at:
>
>   [Anonymous Link]https://pic1.zhimg.com/80/v2-78bc54bc3453588342fd05b3d4b5cf36.png and [Anonymous Link]https://pic1.zhimg.com/80/v2-4c49999ca18b5d8a4df94131ad5bb759.png
>
>   In the visualization, the 'Reconstruction' denotes the results of our method, while the 'Coeffs map' denotes the predicted compensation coefficients for the corresponding scenes. It is important to note that the SPAD captures scattered photons from the entire scene at each sampling point, the predicted coefficients are globally optimized values.
>
>   It is worth mentioning that, since the SPAD captures scattered photons from the entire scene at each sampling point, the predicted coefficients are globally optimized values, which may not exhibit pixel-wise correspondence.
>
>   The figure from the first link demonstrates that LPC assigns larger compensation coefficients to the object farther away (i.e., the statue on the left) in the scene, compared to the closer object (i.e., the deer on the right). For single-object scenes from the figure in the second link, LPC assigns compensation coefficients corresponding to the material properties of the objects (with the 'man' scene containing diffuse material, and the 'ladder' scene containing retro-reflective material) in the areas where the objects are placed. Since an end-to-end network tends to learn and converge with the goal of optimizing the final output (reconstructed image and depth), the intermediate results in visualizations may not perfectly align with the theoretical values (quadratic or quartic). However, we can observe that the compensation coefficients indeed vary according to different materials.
>
> **Q2: Visualization of the standard deviations from APF module.**
>
> * The visualization of standard deviation predictions can be assessed at:
>
>   [Anonymous Link]https://picx.zhimg.com/80/v2-15039835e627e26b62ef2ceb7f5f0e92.png
>
>   For this evaluation, we randomly select 10 transient measurements from the Seen test set for evaluation. Different colored lines in the figure illustrate how the predicted standard deviation varies under different SNR conditions. A lower standard deviation of the Gaussian window in the frequency domain typically indicates a narrower frequency response, leading to stronger high-frequency noise suppression. As shown in the figure, the standard deviation exhibits a decreasing trend as the SNR decreases, which means APF applies a 'tighter Gaussian' for spectrum filtering in lower SNR conditions as the reviewer expected.
>
> **Minor:**
>
> **Q1: Inverse Fourier Transform in Eq.(8).**
>
> * Eq. (8) is implemented as a time-domain convolution in the actual computation process, without NO Inverse Fourier transformation. The Fourier equation in Eq. (8) is provided solely to explain APF from a spectral perspective further. Thanks for pointing this out, we will include additional explanations in the manuscript.
>
> **Q2: The way of capture the real data at varying SNR levels.**
>
> * The qualitative results, which are from the same scene under different exposure times, are presented in Fig.10 and Fig.11 of Appendix A. For the SNR conditions of real-world transient measurements, different exposure times result in different SNR levels, as the reviewer expected. It can be seen that our method achieves the best performance under different exposure times, highlighting its superior generalization capability.

---

> > ### Comment · Reviewer_M5tk · 2024-11-26
> > **Response**
> >
> > I thank the reviewers for the work they did to address my comments about the physical interpretation of their method. I think the visualizations are reasonable and align with what I expect. Not having pixelwise correspondence for the adaptive coefficients looks a bit odd, but the authors provide a reasonable explanation that I also hope can be put in the paper somewhere, if even the supplemental material.
> >
> > I will upgrade my score to a 6.

---

> > > ### Author Response · Authors · 2024-11-26
> > > **Official Comment by Authors**
> > >
> > > Thank you for raising the score.
> > >
> > > We are encouraged that our response has effectively addressed your concerns. Following your suggestion, we have included Q1 and Q2 in the Appendix of the revised manuscript. We sincerely appreciate your thorough review and the valuable time you've taken to helping us improve our work.

---

### Official Review · Reviewer_RyBZ · 2024-11-04

**Soundness:** 3
**Presentation:** 3
**Contribution:** 3
**Rating:** 6
**Confidence:** 4

**Summary:**

The paper proposes to learn the physical priors for  non-line-of-sight imaging. More specifically two components are proposed, learnable path compensation to attenuate the features to compensate for radiometric falloff and an adaptive bandpass filter to suppress background noise. Training and quantitative evaluations are done on synthetic data. Qualitative evaluations are done on real-world data including a three-scene dataset captured by the authors. The method shows better performance in terms of preserving sharp features and reducing background noise.

**Strengths:**

Typically training on synthetic data does not generalize to real data due to synthetic and real gap. However, the proposed method learns to compensate for radiometric falloff and suppress background noise using synthetic training data and is able to generalize to real-world data.

**Weaknesses:**

The paper lacks discussion on the choice of G_z for initial compensation. It’s also unclear how much the 3D CNN block is contributing compared to the initially compensated features.

The paper shows results on a custom dataset with the “composite” scene consisting of multiple surface materials. However, there is no quantitative evaluation on how different surface materials at different distances are compensated by LPC.

**Questions:**

How much does the initial compensated feature in Eq. 5 contribute to the method?

What is the motivation behind average pool as opposed to max pool which would pick the prominent feature?

For the LPC pipeline, how much is the 3D CNN contributing as opposed to the initially compensated feature?

What is the network architecture of spectrum convolution in Figure 4?

How many different materials were used in the captured scene?

---

> ### Author Response · Authors · 2024-11-22
> **Official Comment by Authors**
>
> Thank you for your valuable time and comments. The main concerns are addressed below.
>
> **Q1: Quantitative evaluation on how different surface materials at different distances are compensated by LPC.**
>
>   * As  suggested, we add visualizations of the predicted compensation coefficients at:
>
>     [Anonymous Link]https://pic1.zhimg.com/80/v2-78bc54bc3453588342fd05b3d4b5cf36.png and [Anonymous Link]https://pic1.zhimg.com/80/v2-4c49999ca18b5d8a4df94131ad5bb759.png
>
>     The figure from the first link shows that LPC assigns larger compensation coefficients to the object farther away (i.e., the statue on the left) in the scene, compared to the closer object (i.e., the deer on the right). For single-object scenes from the figure in the second link, LPC assigns compensation coefficients corresponding to the material properties of the objects (with the 'man' scene containing diffuse material, and the 'ladder' scene containing retro-reflective material). It is worth mentioning that, since the SPAD captures scattered photons from the entire scene at each sampling point, the predicted coefficients are globally optimized values, which may not exhibit pixel-wise correspondence.
>
> **Q2: Contributions of the initial compensated feature in LPC pipeline.**
>
>   * To investigate the contribution of the initial compensation to LPC, we remove the initial compensation coefficients, as expressed by:
>
>     $F_{C}^{ini} = \( (G_{Z}), (G_{Z}), (G_{Z}) \) \otimes F_{E}^{'}$.
>
>     The modified approach is denoted as 'Ours-w/o IC'. The quantitative results under the same testing conditions as Tab. 1 are shown below:
>
>     | Method      | PSNR      | SSIM       | RMSE       | MAD        |
>     | ----------- | --------- | ---------- | ---------- | ---------- |
>     | Ours-w/o IC | 23.86     | 0.8636     | 0.0903     | 0.0398     |
>     | Ours        | **23.99** | **0.8703** | **0.0874** | **0.0312** |
>
>     These results demonstrate that incorporating initial compensation effectively enhances performance. To further exhibit the contribution in real-world data, we have provided a qualitative comparison of reconstructed results, which can be assessed in:
>
>     [Anonymous Link]https://pic1.zhimg.com/80/v2-f2e5a87ba02a3bd1240266d730239a8f.png
>
>     It can be observed that our approach reconstructs richer scene details compared to 'Ours-w/o IC' such as the plaster statue in the rear right of the 'teaser'.
>
> **Q3: Motivation for using average pooling instead of max pooling.**
>
>   * Thanks for your valuable comment. Average pooling was originally chosen to capture the overall feature distribution, ensuring the method considers the broader context. As suggested, we have tested max pooling and observed a slight improvement on performance. We greatly appreciate this insight and plan to incorporate this update in the revised paper.
>
> **Q4: Contributions of the 3D CNN block in LPC pipeline.**
>
>   * The 3D CNN block is placed within the LPC module to learn compensation features, and removing this block would make the LPC module non-functional. However, it is important to emphasize that the 3D CNN block is not the core contribution of the LPC module. While modifications to this block could potentially enhance network performance, such exploration falls beyond the scope of this paper.
>
> **Q5: The architecture of spectrum convolution.**
>
>   * Firstly, the features of the transient measurements are transformed into the frequency domain using a Fourier transform. Subsequently, a series of 3D convolutional layers with a stride of (1, 2, 2) is applied to reduce the spatial dimensions of the feature vectors, with each convolutional layer followed by a ReLU activation layer. The extracted features are then processed using a 1D convolutional layer, normalized with LayerNorm, and passed through a fully connected layer to fuse the spectrum dimension into a scalar value. Finally, a sigmoid activation computes the desired standard deviation. We will include the detailed descriptions of the network architecture in the revised paper.
>
> **Q6: Materials used in the captured scene.**
>
>   * As shown in Fig. 7 and Fig. 8 of the manuscript, the material types of the various objects can be divided into two groups: retro-reflective type (e.g., a bookshelf, a dragon model, and the letters 'NYLT' covered with reflective tape) and diffuse type (e.g., the number '2' covered with white printer paper, a cardboard printed with '123XYZ', and the plaster statues).

---

### Author Response · Authors · 2024-11-25
**General Response by Authors**

We sincerely thank the reviewers for their valuable comments and suggestions, and we hope our responses adequately address your concerns. The revised version of the manuscript has been uploaded. Additionally, we are happy to provide further details on any aspects of our responses that may require additional clarification or elaboration.

Once again, we appreciate the reviewers’ time and insightful feedback and look forward to receiving further input.

---

### Author Response · Authors · 2024-12-04
**General Response by Authors**

We have carefully incorporated each reviewer’s constructive comments, making corresponding revisions and clarifying possible misunderstandings. We are encouraged to see that Reviewer M5tk have raised the score during discussion. Although not all reviewers have responded to our latest replies, we believe the major concerns should be adequately addressed.

We hope our efforts will be taken into consideration in your final justification. Once again, we sincerely thank all the reviewers for their valuable time and feedback.

---

### Meta-Review · Area_Chair_QqRj · 2024-12-24

**Metareview:**

This paper aims to enhance non-line-of-sight (NLOS) imaging performance through active time-transient imaging with SPAD. The authors argue that traditional methods rely on empirical physical priors, which lack generalization abilities. They demonstrated improved results with both "seen" and "unseen" data compared to existing methods. However, two reviewers (M5tk and iqtw) raised concerns regarding the physical interpretation and the actual reasons behind the performance improvements. After extensive discussion, the authors acknowledged that some aspects of their presentation were improper, while the reviewers were partially persuaded. Reviewer iqtw also highlighted the marginal improvement over NLOST. In response, the authors explained the differences in performance based on the network structure and parameters (UNet vs. Transformer).

The AC conducted a detailed review of the performance gains over NLOST. It was observed that for all "seen" cases, the improvement is marginal. In the "unseen" cases with high SNR, the performance gains are similarly minor. These findings undermine the authors' claims regarding the superiority of their method in terms of physical priors and generalization ability. Furthermore, the authors' feedback to Reviewer iqtw about performance gains over NLOST heightened the AC’s concerns, as the paper's motivation did not initially include any discussion on network structure or parameter differences (UNet vs. Transformer). It would have been straightforward for the authors to isolate the effect of the network used by employing the same architecture. Additionally, the learned coefficient maps shown in Fig. 12 do not align well with the scene structure, which weakens the authors' claims to some extent.

Given the concerns regarding the physical interpretation, the explanation of performance improvements, and the lack of sufficient justification for the paper's motivation, the AC finds the foundation of the work insufficiently supported. The final recommendations from the four reviewers are split, with 2 marginal accepts and 2 marginal rejects. Considering the concerns outlined above, the paper’s key motivation and the principled interpretation of the underlying physics need to be revisited and validated with experimental results. Thus, the AC recommends rejecting this paper in its current form.

**Additional Comments On Reviewer Discussion:**

Refer to the meta-review above.

---

### Decision · Program_Chairs · 2025-01-22

Reject